# Research on Automatic Error Data Recognition Method for Structured Light System Based on Residual Neural Network

Aozhuo Ding, Qi Xue *, Xulong Ding, Xiaohong Sun, Xiaonan Yang and Huiying Ye

School of Electrical and Information Engineering, Zhengzhou University, No. 100, Kexue Avenue, Zhengzhou 450001, China
* Correspondence: ieqxue@zzu.edu.cn; Tel.: +86-18838018062

**Abstract:** In a structured light system, the positioning accuracy of the stripe is one of the determinants of measurement accuracy. However, the quality of the structured light stripe is reduced by noise, object shape, color, etc. The positioning accuracy of the low-quality stripe center will be decreased, and the large error will be introduced into measurement results, which can only be recognized by a human. To address this problem, this paper proposes a method to identify data with relatively large errors in 3D measurement results by evaluating the quality of the grayscale distribution of stripes. In this method, the undegraded and degraded stripe images are captured. Then, the residual neural network is trained using the grayscale distribution of the two types of stripes. The captured stripes are classified by the trained model. Finally, the data corresponding to the degraded stripes, which correspond to large errors in the data, can be identified according to the classified results. The experiment shows that the algorithm proposed in this paper can effectively identify the data with large errors automatically.

**Keywords:** structured light; stripe grayscale distribution; large error data recognition; deep learning

## 1. Introduction

The structured light 3D measurement method is one of the typical 3D measurement methods based on the triangulation principle. Since it has the advantages of no contact, high efficiency, high accuracy, and low cost [1], it is widely applied in many fields, such as 3D measurement, industrial production, quality inspection, and other areas [2–10]. Among a multitude of techniques based on structured light 3D measurement, stripe-coded structured light plays an essential role due to its high attainable measurement accuracy [11]. In the method, the coded structured light stripe patterns (as shown in Figure 1) are first projected onto the object to be measured by a projector, and then the 3D information of the object is calculated according to the modulated stripe patterns [12]. 3D coordinates are obtained based the locations of stripes, which are located according to the gray distribution of the stripes.



**Figure 1.** Structured Light Stripe Pattern with Encoded Information.

However, many factors in the measurement process degenerate the distribution of stripe and reduce the measurement accuracy of the system. For example, when the shape of the object to be measured is complex, the stripe pattern projected by the camera on

the surface of the object can be severely distorted, and a significant impact on the stripe center extraction is caused. The color of the measured object surface may cause an uneven grayscale distribution of the structured light image [13]. In addition, noise is introduced by the components of the measurement system as well as the measurement environment. The noise can blur the stripes and degrade the image quality. All of the above factors may lead to the degradation of pattern features, generating large errors and resulting in the degradation of the accuracy of the structured light measurement system.

In order to improve the accuracy of structured light stripe center extraction, scholars have conducted a lot of related research and achieved many results.

In 1998, Stegers [14] proposed a method to extract the subpixel location of the center and width of a stripe with high accuracy by analyzing the stripe cross-sections that produce asymmetric degradation. This approach used a linear model for extracting stripe centers, which used first- and second-order directional derivatives to extract the center line of the stripe image and reduced the effect of bias caused by asymmetric degradation. Huang [15] proposed an algorithm combining the threshold iterative extremum method with the weighted gray center of gravity method. The algorithm requires the object to be placed within a range of 400 mm to 700 mm in front of the measurement system. The algorithm is efficient in extraction and has the advantage of being adaptable to the measurement of different objects. Aiming at the problem of low accuracy in locating the center of stripes on highlight surfaces, Wang [16] proposed a stripe center locating method with high robustness. This method uses the grayscale center of gravity method to obtain the coarse center of the stripes. Then, the Sobel operator is used to obtain the normal direction field of the stripes. The center of the stripes in the normal direction is then determined using the grayscale center of gravity method, and the sub-pixel level pixel coordinates are obtained using bilinear interpolation. This can eliminate the interference of flash point noise effectively with strong robustness. Xi [17] eliminated the light bar noise through a two-step convolution process and took a median filter to eliminate the random noise in the background. This method searches for the nearest neighboring points near the normal after identifying the normal equation of each point along the edge of the stripe in a small neighborhood. The sub-pixel coordinates of the stripe center are then extracted using the center of gravity method based on the grayscale values of these points. The simulation results prove the effectiveness of the method. To eliminate stripe distortion caused by uneven reflectivity and large curvature changes on the surface of the object, Zhang [18] proposed a stripe preprocessing method to adjust the stripe distortion grayscale. Additionally, the center of the stripes is extracted by the curve-fitting method of preprocessing. The integrity and reliability of stripe center extraction are ensured for the measurement of surfaces with large curvature variations.

The stripe center locating methods mentioned above optimize the center extraction accuracy. However, when the distribution of the stripe degenerates, it is difficult for the improved center locating methods to handle the measurement results with large error due to the serious degeneration. The large error data is typically recognized manually, which is not objective. Therefore, an automatic objective method for recognizing large errors in data is needed. Ji [19] proposed a method for evaluating the quality of structured light stripe images. This method evaluates stripe quality based on the third-order center distance and the Human Vision System (HVS) without reference. It can recognize large errors in data by setting thresholds of evaluated results. This method needs two steps, which are calculating the evaluation parameters and setting a threshold for the parameters. Therefore, it is complex and not fast enough.

Deep neural network algorithms have achieved remarkable results in many engineering fields in recent years [20–23], and residual neural networks (ResNet) have been increasingly used in detection tasks. Huang et al. used deep ResNets for defect detection in reinforced concrete to achieve the intelligent classification of images of single concrete defects [24]. Qian et al. proposed a ResNet classifier applied to bearing fault detection, and the trained classifier can achieve 100% accuracy [25]. Fan et al. proposed a parallel

ResNet-based method for the automatic detection and measurement of pavement cracks and demonstrated the reliability of the method through experiments [26].

From the above analysis, it can be seen that the traditional method of image processing-based structured optical stripe image quality evaluation requires the manual selection and adjustment of threshold parameters, which is complex and non-objective. Therefore, in order to automatically and objectively identify large error data, this proposed a solution, the flow chart of which is shown in Figure 2. Section 2 of this paper introduces the basic principles of structured light measurement systems and residual neural networks. The evaluation criteria for defective and non-defective stripes are given in Section 3.1. Section 3.2 introduces the method of establishing the datasets. Section 4.1 describes the building of the structured light measurement system. Section 4.2 presents the details of the parameters during the training of the residual neural network and shows the training results. Section 4.3 describes the testing and evaluation of this method, and also compares this method with conventional methods. Finally, a conclusion is given in Section 5.

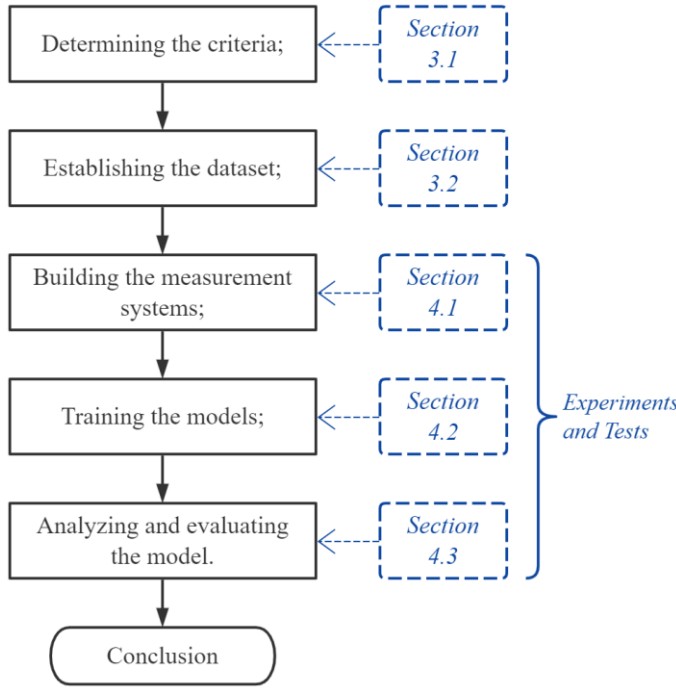

**Figure 2.** Flow Chart of the Solution.

## 2. Principles of Structured Light Measurement and Residual Neural Networks

### 2.1. Interference Factors in Structured Light Measurement Systems

The structured light method is based on the triangulation principle. As shown in Figure 3, the system generally consists of a camera, a projector, and a computer. The coded patterns are projected onto the object, and the patterns modulated by the object are captured by a camera [27]. Based on the triangulation principle, the 3D information can be calculated according to the modulated patterns in the computer. There are three types of structured light patterns, which are point, line, and surface structured light. The measurement steps of the structured light system are shown in Figure 4.

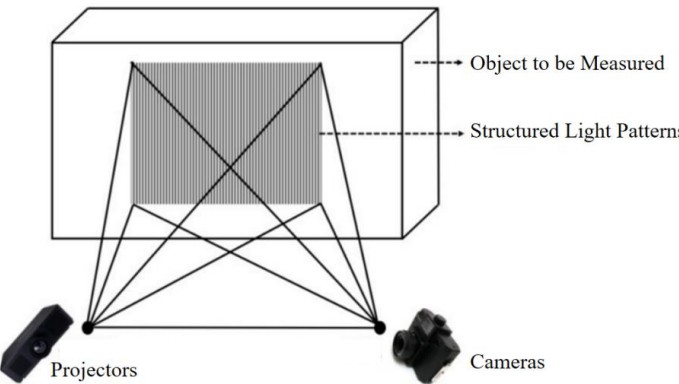

**Figure 3.** Surface Structured Light 3D Measurement.

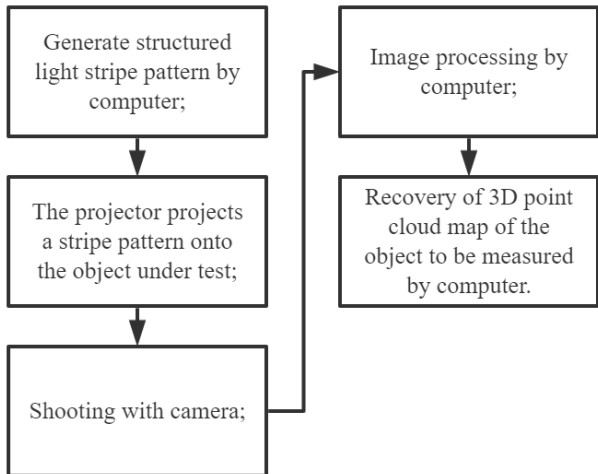

**Figure 4.** 3D Measurement Experiment Process.

The projection and shooting of the surface structured light image are shown in Figure 3, the image containing the three-dimensional information of the object under test is captured and input into the computer for data processing, and, finally, the three-dimensional information of the object under test is obtained.

Since the 3D information is obtained from the modulated stripes, the location of the stripes in the image is very important for the accuracy of measurement. Most of the stripe-centering methods are based on the stripe grayscale distribution. However, structured light stripes and experimental environments are complex and diverse. Many factors, such as image noise, color, and shape of the object to be measured; projecting angle; and shooting angle, degenerate the stripe gray distribution, which in turn decreases the accuracy of stripe centering [28]. As shown in Figure 5, the factors induce a low signal-to-noise ratio and asymmetric distribution of stripe grayscale, which decrease the accuracy of stripe locating. The large error data is introduced into the measurement results.

## 2.2. Residual Neural Network

Compared with the traditional convolutional neural network [29–32], the residual neural network [33] (ResNet) proposed by K. He and his team can effectively solve the problem of the gradient disappearance and gradient explosion [34] of the deep neural network. Furthermore, it has the advantages of deeper layers, a smaller number of parameters, faster convergence, and higher accuracy. The ResNet team constructed building blocks with "shortcut connections", as shown in Figure 6. These basic block units are connected in series to form residual modules, and multiple residual modules form the smallest unit of the residual neural network model.

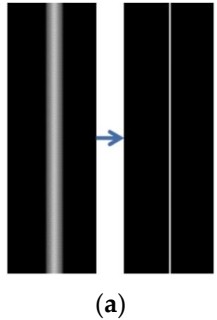 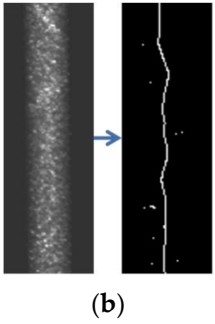 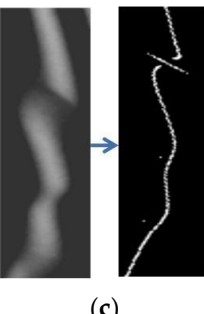

(**a**) (**b**) (**c**)

**Figure 5.** Effect of Interference Factors on Structured Light Stripe Centers. (**a**) Undisturbed Stripes; (**b**) Noise Disturbed Stripes; (**c**) Stripes Disturbed by Object Surface.

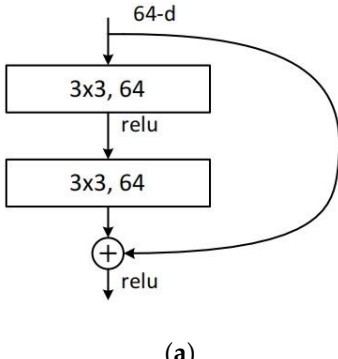 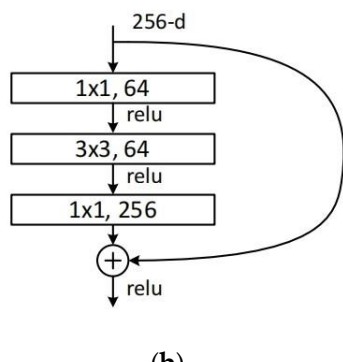

(**a**) (**b**)

**Figure 6.** Residual Neural Network Building Block. (**a**) ResNet-34 Building Block; (**b**) ResNet-50/101/152 Building Block.

The models of the residual neural network composed of the above small cells have slightly different cell structures depending on the number of network layers. There are five model structures: 18-layer, 34-layer, 50-layer, 101-layer, and 152-layer, and the specific network structure configurations are shown in Table 1.

**Table 1.** The Five Models of Residual Neural Network Model.

| Layer Name | Output Size | 18-Layer | 34-Layer | 50-Layer | 101-Layer | 152-Layer |
|---|---|---|---|---|---|---|
| conv1 | $112 \times 112$ | $7 \times 7 \times 64$ | $7 \times 7 \times 64$ | $7 \times 7 \times 64$ | $7 \times 7 \times 64$ | $7 \times 7 \times 64$ |
| conv2_x | $56 \times 56$ | $\begin{bmatrix} 3 \times 3 \times 64 \\ 3 \times 3 \times 64 \end{bmatrix} \times 2$ | $\begin{bmatrix} 3 \times 3 \times 64 \\ 3 \times 3 \times 64 \end{bmatrix} \times 3$ | $\begin{bmatrix} 1 \times 1 \times 64 \\ 3 \times 3 \times 64 \\ 1 \times 1 \times 256 \end{bmatrix} \times 3$ | $\begin{bmatrix} 1 \times 1 \times 64 \\ 3 \times 3 \times 64 \\ 1 \times 1 \times 256 \end{bmatrix} \times 3$ | $\begin{bmatrix} 1 \times 1 \times 64 \\ 3 \times 3 \times 64 \\ 1 \times 1 \times 256 \end{bmatrix} \times 3$ |
| Conv3_x | $28 \times 28$ | $\begin{bmatrix} 3 \times 3 \times 128 \\ 3 \times 3 \times 128 \end{bmatrix} \times 2$ | $\begin{bmatrix} 3 \times 3 \times 128 \\ 3 \times 3 \times 128 \end{bmatrix} \times 4$ | $\begin{bmatrix} 1 \times 1 \times 128 \\ 3 \times 3 \times 128 \\ 1 \times 1 \times 512 \end{bmatrix} \times 4$ | $\begin{bmatrix} 1 \times 1 \times 128 \\ 3 \times 3 \times 128 \\ 1 \times 1 \times 512 \end{bmatrix} \times 4$ | $\begin{bmatrix} 1 \times 1 \times 128 \\ 3 \times 3 \times 128 \\ 1 \times 1 \times 512 \end{bmatrix} \times 8$ |
| Conv4_x | $14 \times 14$ | $\begin{bmatrix} 3 \times 3 \times 256 \\ 3 \times 3 \times 256 \end{bmatrix} \times 2$ | $\begin{bmatrix} 3 \times 3 \times 256 \\ 3 \times 3 \times 256 \end{bmatrix} \times 6$ | $\begin{bmatrix} 1 \times 1 \times 256 \\ 3 \times 3 \times 256 \\ 1 \times 1 \times 1024 \end{bmatrix} \times 6$ | $\begin{bmatrix} 1 \times 1 \times 256 \\ 3 \times 3 \times 256 \\ 1 \times 1 \times 1024 \end{bmatrix} \times 23$ | $\begin{bmatrix} 1 \times 1 \times 256 \\ 3 \times 3 \times 256 \\ 1 \times 1 \times 1024 \end{bmatrix} \times 36$ |
| Conv5_x | $7 \times 7$ | $\begin{bmatrix} 3 \times 3 \times 512 \\ 3 \times 3 \times 512 \end{bmatrix} \times 2$ | $\begin{bmatrix} 3 \times 3 \times 512 \\ 3 \times 3 \times 512 \end{bmatrix} \times 3$ | $\begin{bmatrix} 1 \times 1 \times 512 \\ 3 \times 3 \times 512 \\ 1 \times 1 \times 2048 \end{bmatrix} \times 3$ | $\begin{bmatrix} 1 \times 1 \times 512 \\ 3 \times 3 \times 512 \\ 1 \times 1 \times 2048 \end{bmatrix} \times 3$ | $\begin{bmatrix} 1 \times 1 \times 512 \\ 3 \times 3 \times 512 \\ 1 \times 1 \times 2048 \end{bmatrix} \times 3$ |

Table 2 shows the performance comparison of the five residual neural network models, where FLOPs stand for floating point operations, which is understood as the amount of computation and can be used to measure the complexity of the model. Top-1 error rate (Top-1 Err.) is used to measure the accuracy of model prediction. The smaller the Top-1 Err. is, the higher the accuracy of the model. It can be seen that the accuracy of the 50-layer ResNet increases substantially compared to the 18-layer and 34-layer ResNets, while the increase in the complexity of the model is not significant. Although the accuracy of the 101-layer and 152-layer ResNets is slightly improved, the model complexity is significantly increased,

which greatly improves the performance requirements of the hardware. Considering model accuracy and efficiency, a 50-layer residual neural network is used as the training model in this paper.

**Table 2.** Performance Comparison of Five Models of Residual Neural Network Model [33].

| Performance | 18-Layer | 34-Layer | 50-Layer | 101-Layer | 152-Layer |
|---|---|---|---|---|---|
| FLOPs | $1.8 \times 10^9$ | $3.6 \times 10^9$ | $3.8 \times 10^9$ | $7.6 \times 10^9$ | $11.3 \times 10^9$ |
| Top-1 Err. on ImageNet Validation Set (%) | 27.88 | 25.03 | 22.85 | 21.75 | 21.43 |

## 3. Methods of Classification and Establishing Data Sets

### 3.1. Evaluation of Structured Light Stripe Quality

In a structured light system, 3D information of the measured object is calculated using the locations of stripes in projected and captured image. The typical stripe center locating methods shown in Table 3 locate the stripe according to the features of stripe grayscale distribution [35,36]. To ensure the reliability of the extraction results, the typical grayscale distributions of the stripe in a projected image pattern are symmetrical [37], which are shown in Figure 7. However, the grayscale distribution of the stripe is affected by various noises and degrades from the symmetric distribution in the ideal state to the asymmetric distribution. The accuracy of the extracted stripe location is also decreased [38]. Therefore, some large errors in data are introduced into the 3D measurement results. Therefore, the grayscale distribution of stripes is one of the key factors of structured light measurement accuracy, and the evaluation of the grayscale distribution of stripes is an important step in recognizing the large errors in data. As shown in Figure 8, the structured light stripe cross-section captured by the camera contains less information. If the captured structured light stripe cross-section is directly used as the data set for training, it is difficult to accurately evaluate the quality of the structured light stripe center. Therefore, the grayscale distribution is used as the basis for collecting sample data.

**Table 3.** Comparison of Several Typical Stripe Center Extraction Methods.

| Method | Advantages | Disadvantages |
|---|---|---|
| Threshold Method | Fast and simple. | Low extraction accuracy. |
| Extreme Value Method | Fast and simple. | Easily disturbed by noise. |
| Curve Fitting Method | Less affected by noise, high accuracy. | Low robustness and slow calculation speed. |
| Steger Algorithm | High accuracy and robustness. | Slow calculation speed. |
| Grayscale Gravity Method | High accuracy, high noise resistance, and high efficiency. | Low sensitivity. |

### 3.2. Sample Data Collection

Based on the analysis in Section 3.1, the gray distribution of stripe is used as the sample. It can not only improve the training accuracy, but also reduce the training difficulty and training time. We divided the cross-sectional grayscale distribution into two categories, corresponding to defect-free and defective stripe sections, as shown in Figure 9. One is the grayscale distribution, which is not degenerated, as shown in Figure 9a,b. The other one is the grayscale distribution degenerated, as shown in Figure 9c–e. Among them, the stripe cross-section in Figure 9c is degenerated due to object surface deformation, the stripe cross-section in Figure 9d is degenerated due to object surface color change, and the stripe cross-section in Figure 9e is degenerated due to specular reflection from the object surface. In addition, there are many other factors, such as noise from the measurement system and

the measurement environment, that can cause degradation of the grayscale distribution of the stripe cross-section.

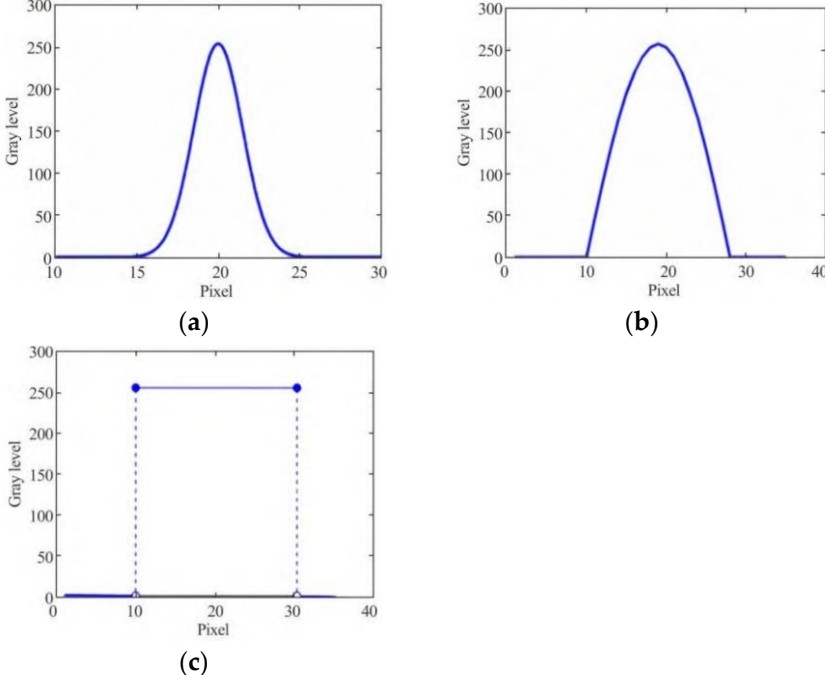

**Figure 7.** Grayscale Distribution of Structured Light Fringes. (**a**) Gaussian Grayscale Distribution; (**b**) Sinusoidal Grayscale Distribution; (**c**) Rectangular-Type Grayscale Distribution.

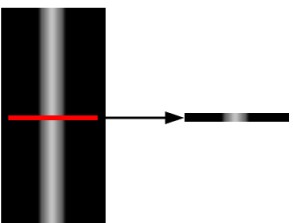

**Figure 8.** Structured Light Stripe Cross-Section.

Objects with different shapes and materials are used, which include ceramic cups, bearing steel balls, flat board, plaster Venus sculptures, plaster balls, etc. The encoded structured light stripe pattern is projected onto the objects. From this, 6450 stripe cross-sectional grayscale distribution images are selected and manually classified into two types of defective and defect-free as the data set, and their composition is shown in Table 4.

**Table 4.** Composition of the Data Set.

| Type | Number of Train Set Images | Number of Test Set Images |
| --- | --- | --- |
| Defective stripes | 3320 | 830 |
| Defect-free stripes | 1840 | 460 |

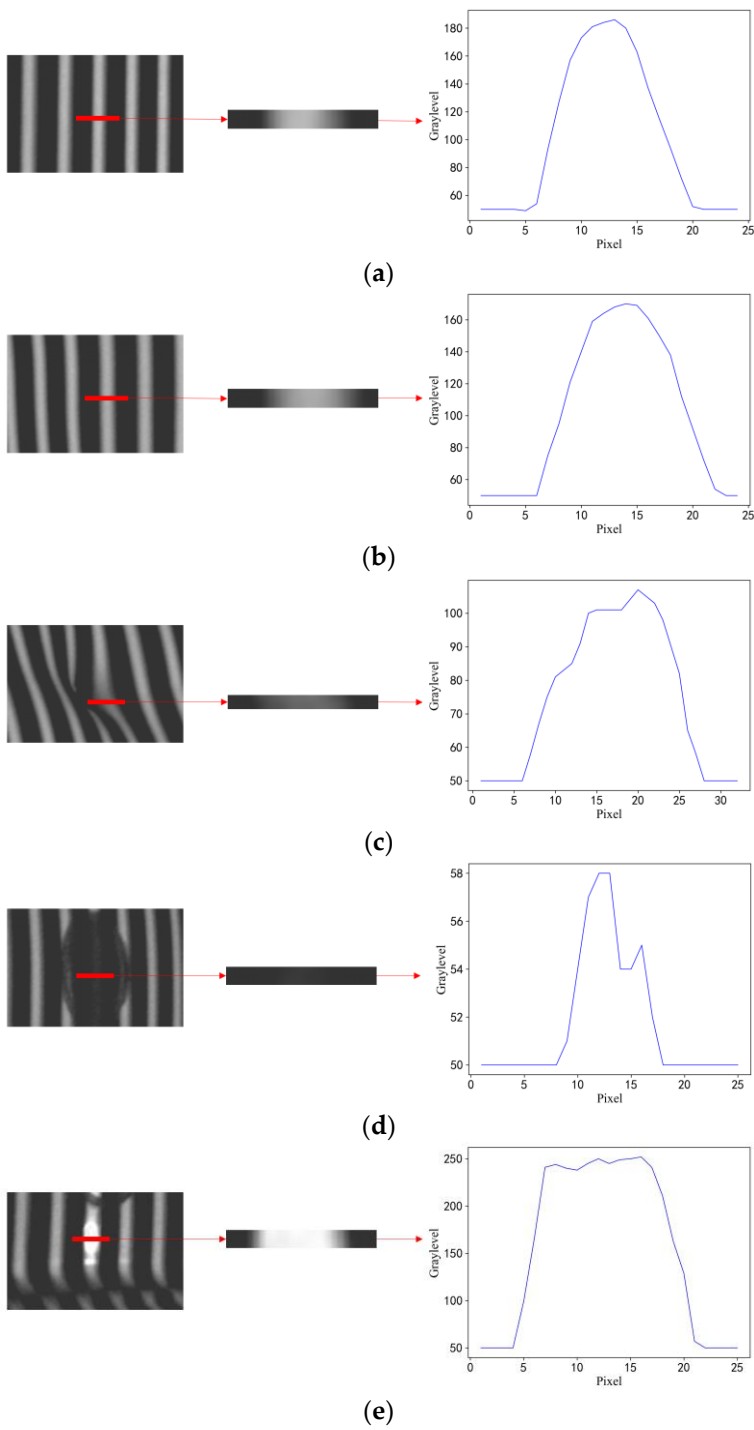

**Figure 9.** Structured Light Stripe Cross-sections and their Grayscale Distribution. (**a**,**b**) Defect-Free Stripes; (**c**–**e**) Defective Stripe.

## 4. Experimental Results and Analysis

The sample data sets are created first. Then, the neural network model is trained using the training data set. Based on the training model, the detection of defect stripe can be carried on. First, a structured light stripe is selected. The structured light stripe cross-section shown in Figure 8 is intercepted from the selected stripe and input into the quality evaluation algorithm. Then, the stripe cross-section is transformed into the grayscale distribution graph shown in Figure 9. Finally, the grayscale distribution curve is input into

the trained neural network model, and the quality evaluation result of the structured light stripe is output. The above quality evaluation steps are summarized in Figure 10.

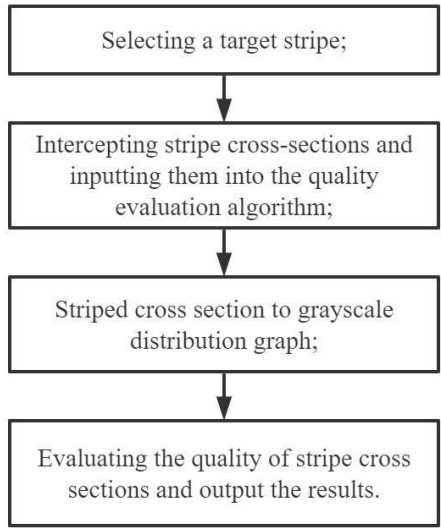

**Figure 10.** The evaluation steps.

### 4.1. Structured Light Experiment System

The experiment was performed for simulation calculations in MATLAB 2018 software on a 64-bit Windows system. The experimental measurement setup includes a projector (Richon PJX2180, Coretronic, made in Jiangsu, China), a camera (Basler acA4600-7gc, Basler, made in Germany CEFC), a computer, a displacement platform, a calibration board, and the object under test. The camera and the projector are responsible for projecting and capturing the structured light patterns. The calibration board is used for calibration [39], and the displacement platform is used to adjust the distance between the measured object and the camera and projector. Figure 11 shows the experimental environment and instruments used in the experiment.

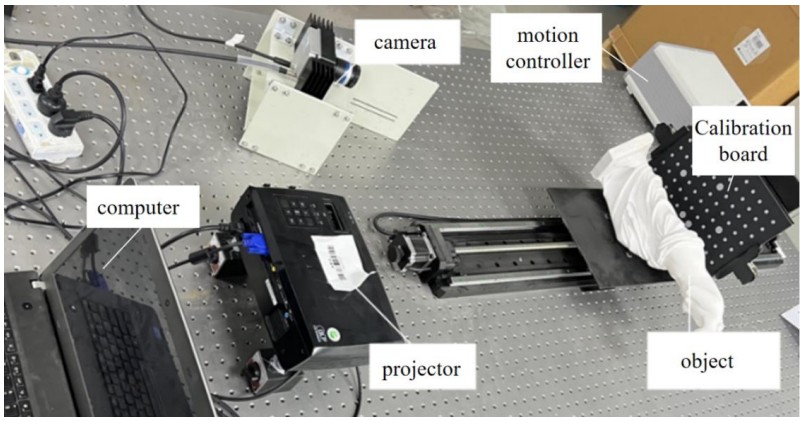

**Figure 11.** Structured light three-dimensional measurement experiment platform.

### 4.2. Model Training

In this paper, we use the Adam optimizer because it combines the advantages of both AdaGrad and RMSProp optimization algorithms, featuring simple implementation, efficient computation, low memory requirements, stable parameters, and an automatic adjustment of learning rate [40]. During the training process, the role of the Adam optimizer is to change the trainable parameters in a way that minimizes the loss function in order to obtain the optimal solution of the model.

In addition, binary cross-entropy is used as a loss function to measure the performance of neural networks. Binary cross-entropy is a commonly used loss function in binary classification tasks, and the expression is shown below.

$$\text{Loss} = -\frac{1}{N} \sum_{i=1}^{N} y_i \log(p(y_i)) + (1 - y_i) \log(1 - p(y_i)) \tag{1}$$

where y is the binary label 0 or 1; p(y) is the probability that the output belongs to label y. As a loss function, the binary cross-entropy is used to judge how well a binary classification model predicts the outcome. For example, for a label y of 1, if the predicted value p(y) tends to 1, then the value of the loss function should tend to 0. Conversely, if the predicted value p(y) tends to 0 at this point, then the value of the loss function should be very large.

The residual neural network model was trained, and the experimental environment was specified as follows.

Operating system: Windows 10
CPU: Intel (R) Core (TM) i5-11300H
RAM: 16. 0 GB

First, to further verify the performance of the residual neural network with different layers in this study, this paper first conducted a test training of the dataset with 34-layer, 50-layer, and 101-layer ResNets, respectively. The initial learning rate of the algorithm and its exponential decay rate were set to $10^{-4}$ and 0.9, respectively. The network input size was $480 \times 320$, the batch size was set to 20, the epoch was set to 15. After 15 generations of training, the performance of different layer residual neural network models was tested and compared, as shown in Table 5.

**Table 5.** Performance Comparison of 34-Layer, 50-Layer, and 101-Layer ResNets in this Study.

| Performance | 34-Layer | 50-Layer | 101-Layer |
|---|---|---|---|
| Top-1 Accuracy on Validation Set (%) | 90.16 | 92.95 | 93.72 |
| Elapsed Time for Each Epoch (s) | 1500 | 2880 | 4035 |

Top-1 accuracy is the accuracy the category ranked first in probability with the actual result. As can be seen in Table 5, the top-1 accuracy of the 50-layer ResNet is significantly improved compared to the 34-layer ResNet. Compared with the 50-layer ResNet, the top-1 accuracy of the 101-layer ResNet is slightly improved. However, its training time consumption is greatly increased. This also confirms the analysis in Table 2 for the performance of different layers of residual neural networks.

Therefore, in this paper, we choose to continue training the model with a 50-layer residual neural network. The initial learning rate of the algorithm and its exponential decay rate were set to $10^{-4}$ and 0.9, respectively. The network input size is $480 \times 320$, the batch size is set to 20, the epoch is set to 60, and the current phase of training is stopped when the loss is no longer decreasing. After testing, this can ensure a better training effect based on this hardware configuration.

As shown in Figure 12, with the loss of the training set and the val loss of the validation set gradually decreasing, the loss tends to stabilize after 60 iterations, and the fluctuation is around 0.124 and 0.121, respectively. It can be seen that the loss at the end of training converged, which means the model achieved a high classification accuracy in the test set.

After testing, the Top-1 accuracy of the neural network model trained in this paper is 95.27%.

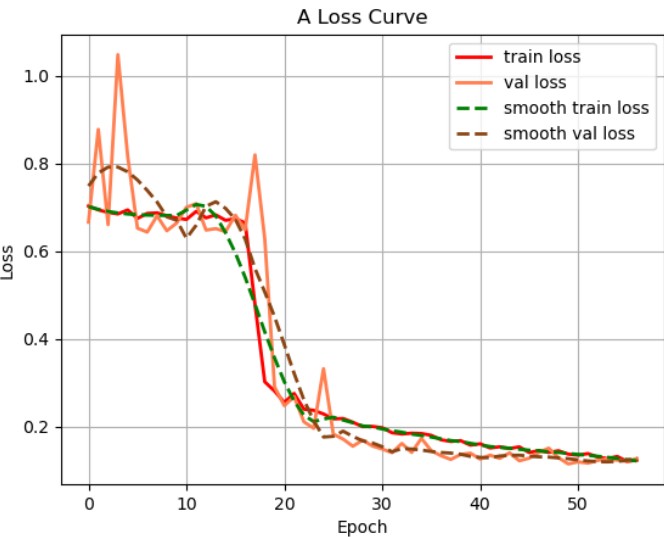

**Figure 12.** Loss Curve.

### 4.3. Testing and Evaluation

To test the performance of the method, a standard sphere with a diameter of 50.80(+1.50/−1.00) mm, a flat board with flatness of 15 μm, and a standard cylinder with a diameter of 68.00 mm were measured by the structured light system. The testing steps of the method are shown in Figure 13. First, the stripe pattern was projected on the subject and photographed, and then the 3D information of the subject was calculated. The structured light stripe cross-section was input into the model trained above to determine whether the stripe was defective or not. Then, the 3D point cloud corresponding to the stripe region with defects was removed. Finally, the measurement errors with and without processing were calculated and compared.

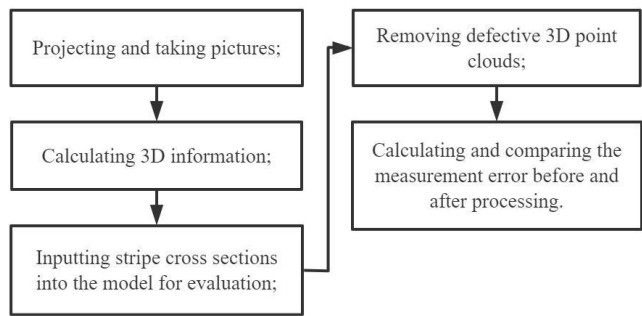

**Figure 13.** Validation Process Flow Chart.

The principle of the method proposed in this paper is to evaluate the stripe quality according to the grayscale distribution of the structured light stripe cross-section. The grayscale distributions in the dataset include the ones of stripes on many different typical materials and shapes of objects in actual measurement processes. In order to test the generalization performance, this paper tested objects such as a steel ball and a flat board which are not included in the training dataset.

The measurement results of the spherical surface are shown in Figure 14. Figure 14a shows the modulated pattern on a standard ball. Figure 14b shows the 3D point cloud of the ball, and Figure 14c,d shows the concentration areas of large errors in the data. The ball in Figure 14b is obtained by fitting the measurement data to a sphere. Although the intensity of Figure 14c is high, the skewness coefficient of the stripe is high because of specular reflection. In Figure 14d, the noise is largely due to the low intensity of the stripe. The measurement error of the ball is defined as the difference of the fitted radius and the

distance between the measurement 3D point and the fitted center of the ball, which can be defined as:

$$Error = |d - d'| \qquad (2)$$

where d is the actual distance from the sphere to the center of the sphere; d' is the distance of each point of the 3D point cloud from the center of the fitted sphere. Since the gray distribution of the structured light stripes cross-section corresponding to points with large measurement errors in the point cloud tends to be degraded, the points with large measurement errors are considered defective.

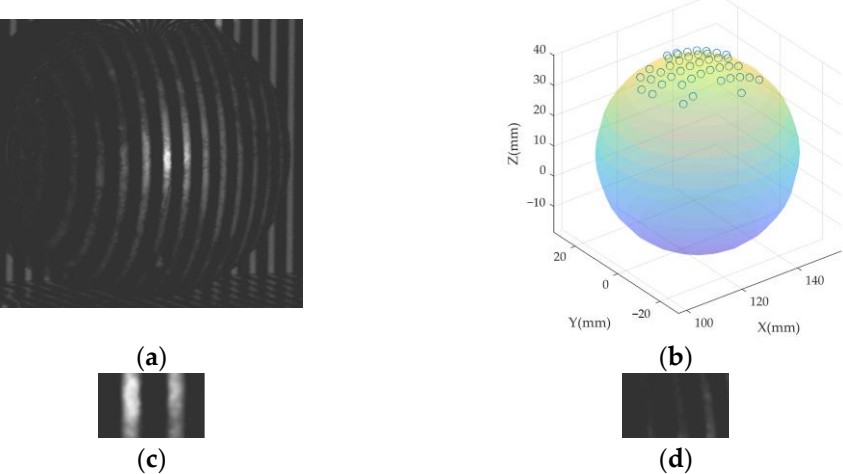

**Figure 14.** Measurement of the Standard Ball. (**a**) Measured Standard Ball; (**b**) 3D Point Cloud; (**c**,**d**) Areas of Captured Image Corresponding to Large Error Points.

The stripe cross-sections are intercepted in Figure 14a and input into the prediction algorithm. The algorithm uses the model trained in Section 4.2 as the prediction model to transform the cross-sectional images into grayscale curves and perform prediction, and the results are shown in Figure 15. The grayscale distribution of the defective stripes in Figure 15a–d comes from the irregular regions in Figure 14c,d, and the grayscale distribution of the defect-free stripes in Figure 15e–h comes from the region with better stripe quality in Figure 14a. According to the prediction results, the stripe cross-sections at the locations corresponding to the irregular regions in Figure 14c,d both have serious defects and are evaluated as "bad" by the prediction algorithm. In addition, the stripe cross-sections intercepted in regions with a good-quality of stripe are all evaluated as defect-free by the prediction algorithm. This is consistent with the conclusions obtained through the criteria for judging the quality of stripe cross-sections in Section 3.2, as expected.

The measurement errors of the points corresponding to each stripe cross-section in Figure 15 in the 3D point cloud are shown in Table 6. It can be seen that during the evaluation of stripe quality, stripe cross-sections evaluated as defective have large measurement errors in their corresponding area. On the contrary, the points corresponding to the defect-free stripe sections tend to have smaller measurement errors. The results verified the prediction results.

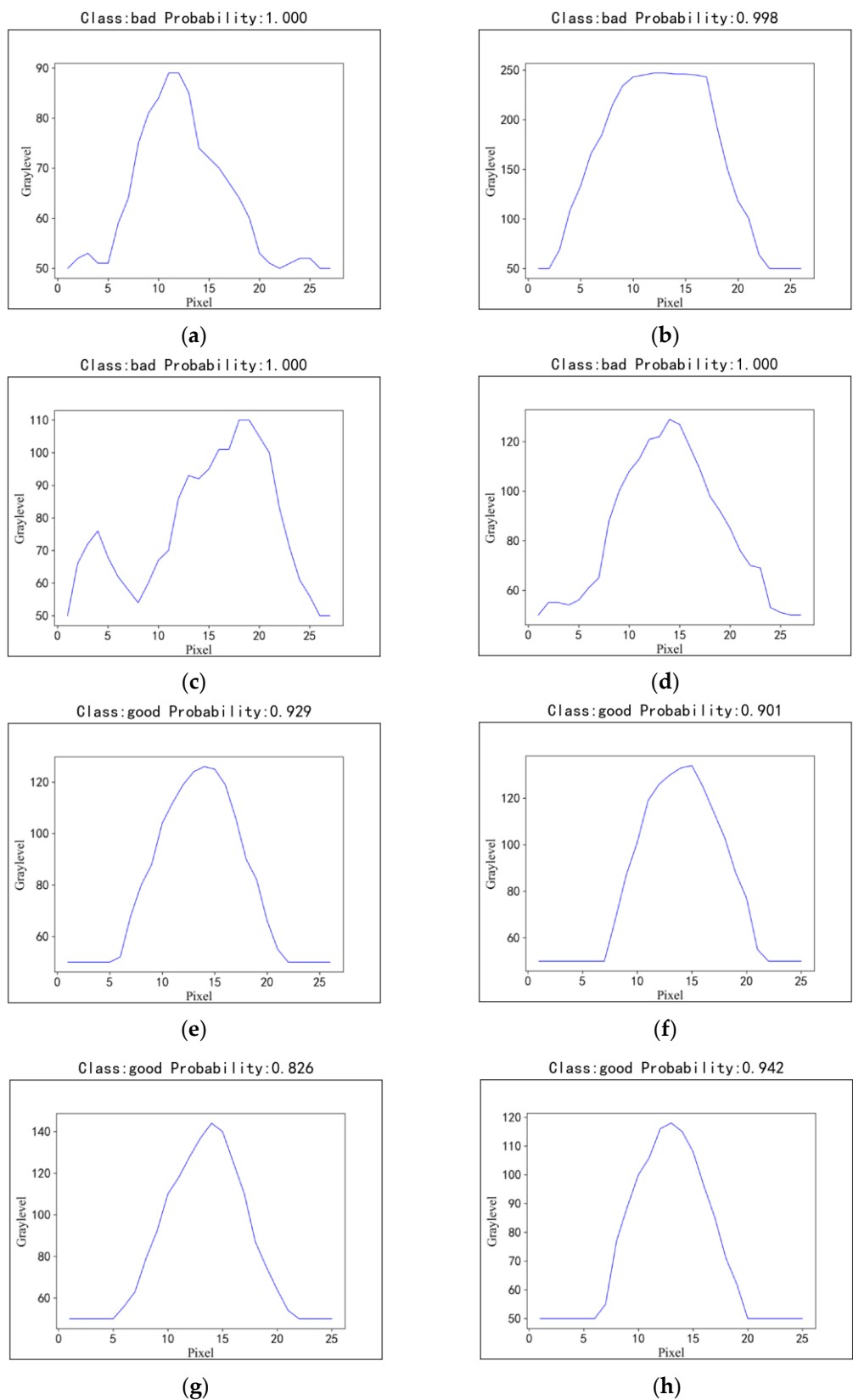

**Figure 15.** Prediction Results of the Standard Ball. (**a**–**d**) Prediction Results of Defective Cross-Section; (**e**–**h**) Prediction Results of Defect-Free Cross-Section.

**Table 6.** Prediction Result Evaluation of Grayscale Distribution in Figure 15.

| Grayscale Distribution in Figure 15. | Measurement Error (mm) | Prediction Result |
|:---:|:---:|:---:|
| (a) | 1.009 | Defective |
| (b) | 1.097 | Defective |
| (c) | 0.914 | Defective |
| (d) | 0.988 | Defective |
| (e) | 0.015 | No Defects |
| (f) | 0.129 | No Defects |
| (g) | 0.094 | No Defects |
| (h) | 0.225 | No Defects |

The quality of the stripes in Figure 14a is examined by the method proposed in this paper and in Ref. [19]. The corresponding defective points are found in the 3D point cloud of Figure 14b and removed, respectively, according to the two methods. The measurement error before and after processing is then calculated for each point in the point cloud according to Equation (2). The measurement errors before and after processing are shown in Table 7. It can be seen that the measurement error of the stripes is significantly reduced after being processed by the method proposed in this paper. Therefore, compared with the method of Ref. [19], the measurement error is smaller after the processing of the method in this paper.

**Table 7.** Measurement Errors of the Standard Ball.

|  | Unprocessed | Processed by the Method in This Paper | Processed by the Method in Ref. [19] |
|:---:|:---:|:---:|:---:|
| Mean/mm | 0.670 | 0.163 | 0.326 |
| Std/mm | 0.786 | 0.106 | 0.264 |
| Max/mm | 1.836 | 0.328 | 1.009 |

For the flat board, Figure 16a,b shows the captured image and 3D points measurement results of flat board. The large error points are mainly distributed at the edges of the board, caused by the asymmetry of the gray distribution of the stripes, as shown in Figure 16a. The stripe cross-sections of the 3D points corresponding to the 2D coordinates in the irregular region are input into the prediction algorithm, and the results are shown in Figure 17. According to the predicted results in Figure 17a,b, the structural light stripe cross-sections at the broken points at the edges of the flat board in Figure 16a are all severely defective and are evaluated as defective by the prediction algorithm. In contrast, according to the predicted results in Figure 17c,d, the structured light stripe cross-sections intercepted at the center of the flat board surface have a uniform grayscale distribution and are evaluated as defect-free by the prediction algorithm. Therefore, the results meet the established expectation.

Similarly, the stripe cross-sections in the edge region of the flat board in Figure 16a are separately input into the prediction algorithm. The points corresponding to the cross-sections with large errors evaluated by the algorithm are removed from the point cloud map shown in Figure 16b. The measurement error information of the edges of the flat board before and after processing are summarized in Table 8. The measurement error is defined as the relative distance of the points on the point cloud from the fitted plane. It can be seen that the measurement error for the flat board is also significantly reduced after processing by the method proposed in this paper.

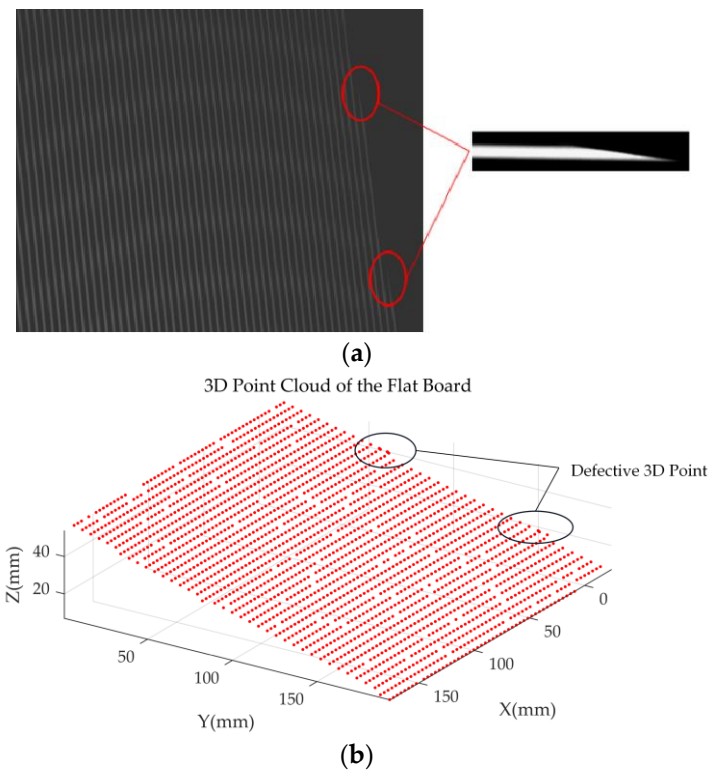

**Figure 16.** Measurement of the Flat Board. (**a**) Measured Flat Board; (**b**) 3D Point Cloud.

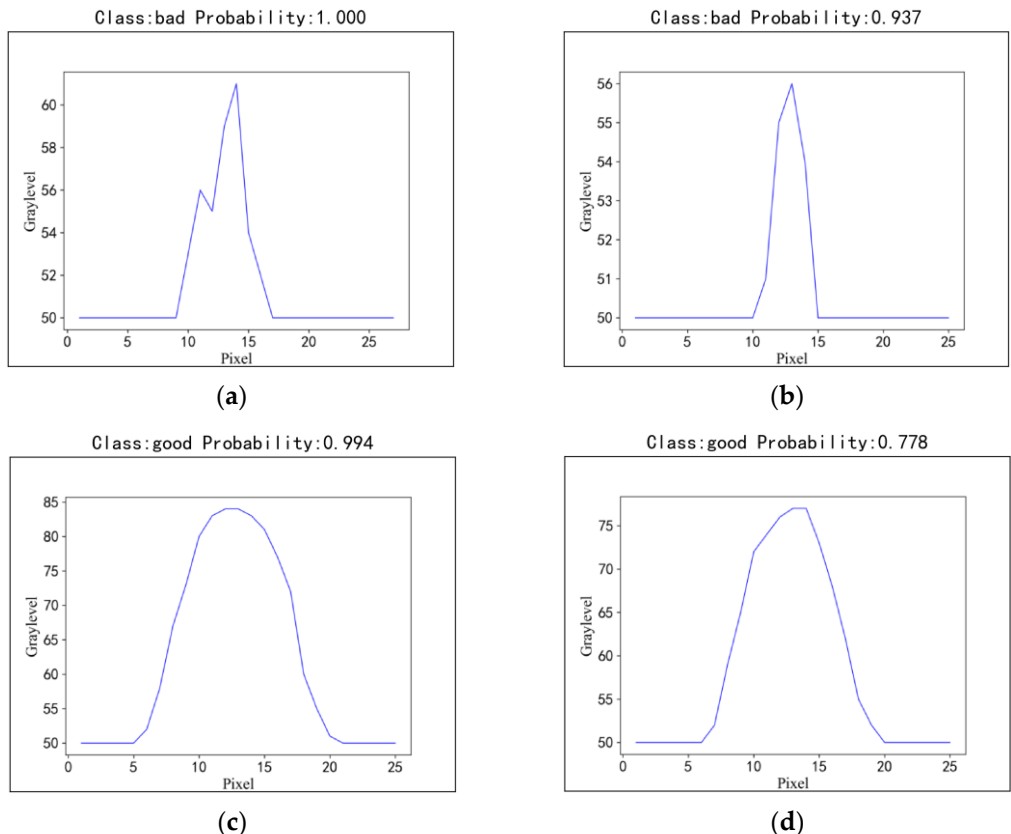

**Figure 17.** Prediction Results of the Flat Board. (**a**,**b**) Prediction Results of Defective Cross-Sections at the Broken Points at the Edges of the Flat Board; (**c**,**d**) Prediction Results of Defect-Free Cross-Section.

**Table 8.** Measurement Errors of the Flat Board.

| | Unprocessed | Processed by the Method in This Paper | Processed by the Method in Ref. [19] |
|---|---|---|---|
| Mean/mm | 0.278 | 0.050 | 0.155 |
| Std/mm | 0.297 | 0.029 | 0.179 |
| Max/mm | 0.968 | 0.091 | 0.530 |

For the cylinder, Figure 18a,b shows the captured images and 3D point measurement results of the cylinder. As shown in Figure 18a, the large error points are mainly distributed in the blackened parts of the cylinder and the edges of the cylinder. The stripe cross-sections intercepted in the irregular region and the regular region in Figure 18a are input into the prediction algorithm separately, and the results are shown in Figure 19. According to the prediction results in Figure 19a,b, both the points at the edge of the cylinder and the cross-sections of the structured light stripes in the blackened part are all severely defective. According to the prediction results in Figure 19c,d, the stripe cross-sections in the regions with more regular stripes have a uniform grayscale distribution and are evaluated as defect-free by the prediction algorithm.

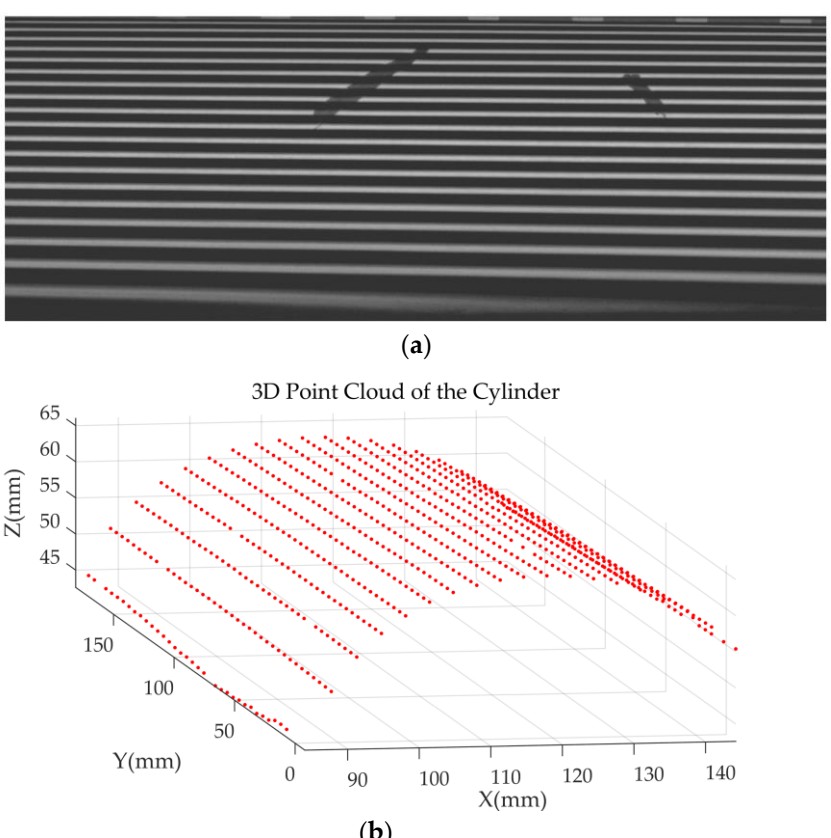

**(a)**

**(b)**

**Figure 18.** Measurement of the Cylinder. (**a**) Stripes on the Cylinder; (**b**) 3D Point Cloud of the Cylinder.

The stripe sections on the cylinder in Figure 18a are intercepted and input into the prediction algorithm separately. The points corresponding to the cross-sections with large errors evaluated by the algorithm are removed from the point cloud map shown in Figure 18b. The measurement error information before and after processing is shown in Table 9. The measurement error is defined as the absolute value of the difference between the distance from the point cloud on the cylinder to the fitted axis and the fitted radius of the cylinder.

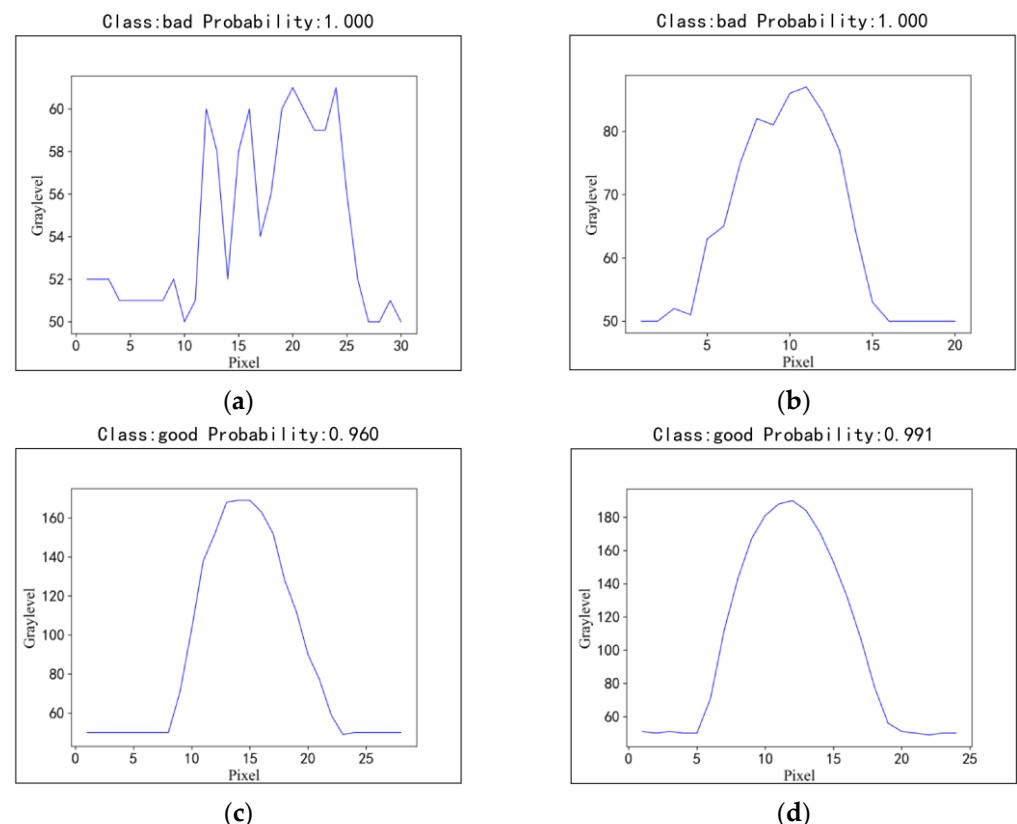

**Figure 19.** Prediction Results of the Cylinder. (**a**) Prediction Results of Defective Cross-Sections at the Edges of the Cylinder; (**b**) Predicted Results of Defective Cross-Sections in Artificially Blackened Parts of Cylinders; (**c**,**d**) Prediction Results of Defect-Free Cross-Section.

**Table 9.** Measurement Errors of the Cylinder.

|  | Unprocessed | Processed by the Method in This Paper | Processed by the Method in Ref. [19] |
|---|---|---|---|
| Mean/mm | 1.455 | 0.817 | 1.345 |
| Std/mm | 1.759 | 0.273 | 0.633 |
| Max/mm | 12.419 | 1.333 | 2.896 |

As can be seen from Tables 7–9, the method proposed in this paper can accurately recognize the large error data. After removing the large error data, the maximum measurement error and the average measurement error of the stripes are reduced significantly. Compared with the method in Ref. [19], the average measurement error is lower after removing large error data using the method proposed in this paper. Since it does not need to calculate the evaluation parameters and set the thresholds of the parameters, it is much faster and more convenient. The comparison between the method proposed in this paper and the method proposed in Ref. [19] is shown in Table 10.

**Table 10.** Comparison of the Two Methods.

| Method | Implementation Steps | Performance |
| --- | --- | --- |
| Structured light stripe defect detection method in Ref. [19] | Setting the threshold; Extracting the gray centers of structured light stripes; Detect the quality of structured light stripes. | Slower processing speed, depending on the thresholds. |
| Structured light stripe defect detection based on residual neural network | Directly detects the quality of structured light stripes. | Higher precision, faster processing speed, no threshold needs to be set. |

## 5. Conclusions

Aimed at the large error data in structured light 3D measurement, this paper proposed a method for recognizing the large errors in the data in structured light 3D measurement results quickly, automatically, and objectively. This method is based on residual neural networks which use the structured light stripe grayscale distribution images as the deep learning dataset. A 50-layer residual neural network is iteratively trained. After several iterations, the model is trained by deep learning converges after several iterations and achieves a high Top-1 accuracy. The test result shows that the trained model can recognize the large error data in measurement results without calculating any parameters and setting any thresholds. After removing the recognized data, the measurement accuracy is improved significantly. Compared with the traditional structured optical stripe quality evaluation method based on image processing, the method based on the residual neural network proposed in this paper does not need to set any threshold, which means it is more convenient and faster. In addition, the method recognizes the large error according to the grayscale distributions, objectively.

**Author Contributions:** Conceptualization, Q.X.; methodology, A.D. and Q.X.; software, A.D.; validation, A.D., Q.X. and X.S.; formal analysis, A.D.; investigation, Q.X. and H.Y.; resources, Q.X., X.Y. and H.Y.; data curation, A.D. and X.D.; writing—original draft preparation, A.D.; writing—review and editing, A.D. and Q.X.; visualization, Q.X. and X.S.; supervision, Q.X.; project administration, Q.X.; funding acquisition, Q.X. and X.Y. All authors have read and agreed to the published version of the manuscript.

**Funding:** This research was funded by Science and Technology Research Program of Henan Province (No. 222102210073), Strategic Priority Research Program of the Chinese Academy of Sciences (XDB44000000), National Natural Science Foundation of China (No. 61705198) and Cangzhou Key research and development Program Guidance Project (No.204103005).

**Institutional Review Board Statement:** Not applicable.

**Informed Consent Statement:** Not applicable.

**Data Availability Statement:** Data can be obtained from the corresponding author upon reasonable request.

**Conflicts of Interest:** The authors declare no conflict of interest.

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
