# Peer review of "Research on Automatic Error Data Recognition Method for Structured Light System Based on Residual Neural Network"

_applsci, doi:10.3390/app13052920_

Round 1
Reviewer 1 Report
This manuscript is well organized and clearly written with a logic description of a method for automatic identification of structured light error data based on residual neural networks.
Based on the manuscript content and its contribution to the field of structured light system, the reviewer would recommend it for being published in Applied Sciences after minor revisions.
Few minor revisions suggested by the Reviewer that:
Some of the curves shown in the article are not clear enough, and it is best to use EPS format pictures.
Author Response
Dear reviewer #1:
Thanks for your suggestion. We have improved the clarity of the curves in Figure 9, 15, 17, and 19, corresponding to Figure 8, 14, 16, and 18 in the un-revised manuscript.
Reviewer 2 Report
The paper proposes a method to identify data with large errors in structured light 3D measurement results. The residual neural network is trained to evaluating the quality of the grayscale distribution of stripes corresponding to the measurement results. The experiments shows that the large error data can be identified accurately. The work is very useful for improving the measurement accuracy in this paper. Therefore, I think this paper could be accepted for publishing. However, there are several problems as follows:
1. In section 2.2, there are five model structures in Table 1. However, it is written as “There are four model structures”.
2. In section 2.2, the residual neural networks with different number of layers are described. I think the authors should give more detail reason that why 50-layer residual neural networks were chosen in this paper.
3. The numbers in figure 8, 14 and 15 are not clearly enough.
4. I suggest that the measurement error in Table 4 should be retained to the third decimal place.
5. Can you show the cross-sectional views of the structured light streaks described in Figure 8 in Section 3.2 for "defect-free" and "defective", and compare them from the grayscale view?
6. How many kinds of object had been used for establishing the training set?
Reviewer 3 Report
This paper proposed a method to recognize the large error data using a trained residual neural network. The method can recognize the error data by assessing the quality of stripe gray distributions. Experiments show that the algorithm works effectively. The work is interesting and valuable. The paper can be accepted after address the following issues.
1. Deos the proposed method can works with the color object?
2. How is the generalization performance of the RNN, please tell the reader as it is important when implementing the proposed method.
3. The calibration method of the structured light system should be given in section 4.
4. The numbers in some figures are too small such as in Figure 8 and Figure 14.
5.The article uses both ''cross sections'' and ''cross-sections'' expressions. I think that the expressions should be unified.
6.The measurement error in Table 4-7 should be round to 3 decimal places.
Reviewer 4 Report
Research on Automatic Error Data Recognition Method for Structured Light System Based on Residual Neural Network
Manuscript Number: applsci-2178800
Comments:
This manuscript is well-written and well-organized. However, the following comments are suggested:
1) Please double-check the writing style by considering punctuation mistakes. For example, “object shape, color etc”.
2) Please mention the criteria to measure the performance of the neural network. For example, the authors should list the R^2 and MSE (or other indexes) for training and validation datasets.
3) Please compare the performance of neural networks when using different models (Table 1).
4) The authors should mention a logical reason for adopting the Adam optimizer and more details on its role.
5) In Table 1, five models have been embedded instead of four models. Please revise it in the text. "There are four model structures"
6) In order to have a comprehensive literature review, please mention the different kinds of neural networks and their applications in various disciplines. For example, https://doi.org/10.1177/14759217211065009 , https://doi.org/10.1177/1369433220947193 , https://doi.org/10.1080/19648189.2021.2003250 , https://doi.org/10.1007/s13349-019-00345-8 , https://doi.org/10.1002/eqe.3775
7) Please plot Figures 8, 14, 16, and 18 using an attractive color.
8) The conclusions section should be enriched by highlighting the manuscript's main findings and contribution.
Reviewer 5 Report
1. The main contribution of the authors is not clearly defined. The authors announce they provide a (new?) method for the automatic identification structure light error data. However, in the paper I can't see a section devoted to that very method. I can't see clearly where the state of the art stops and where the original solution of the authors starts.
2. Technically: the sections titles e.g. "Principle", "Methods" should be extended to be more informative.
3. The authors should revise the presentation of the paper. That applies to experiments results particularly. The reader is overwhelmed with a huge numbers of Matlab-generated figures from which only few are referred and explained in detail.
4. Also conclusion section should be extended. The authors should provide their concluding thoughts. Not just summary of the experiments (what has been done).
Round 2
Reviewer 4 Report
The paper is acceptable.